# Essential Role of Multi-Omics Approaches in the Study of Retinal Vascular Diseases

**DOI:** 10.3390/cells12010103

**Published:** 2022-12-26

**Authors:** Yi Lei, Ju Guo, Shikun He, Hua Yan

**Affiliations:** 1Department of Ophthalmology, Tianjin Medical University General Hospital, Tianjin 300052, China; 2Laboratory of Molecular Ophthalmology, Tianjin Medical University, Tianjin 300070, China; 3Tianjin Key Laboratory of Ocular Trauma, Tianjin Medical University, Tianjin 300070, China; 4Department of Pathology and Ophthalmology, USC Roski Eye Institute, Keck School of Medicine, University of Southern California, Los Angeles, CA 90033, USA

**Keywords:** multi-omics, retinal vascular diseases, biomarker, therapeutic strategies, methodologies

## Abstract

Retinal vascular disease is a highly prevalent vision-threatening ocular disease in the global population; however, its exact mechanism remains unclear. The expansion of omics technologies has revolutionized a new medical research methodology that combines multiple omics data derived from the same patients to generate multi-dimensional and multi-evidence-supported holistic inferences, providing unprecedented opportunities to elucidate the information flow of complex multi-factorial diseases. In this review, we summarize the applications of multi-omics technology to further elucidate the pathogenesis and complex molecular mechanisms underlying retinal vascular diseases. Moreover, we proposed multi-omics-based biomarker and therapeutic strategy discovery methodologies to optimize clinical and basic medicinal research approaches to retinal vascular diseases. Finally, the opportunities, current challenges, and future prospects of multi-omics analyses in retinal vascular disease studies are discussed in detail.

## 1. Introduction

Retinal vascular diseases constitute highly prevalent ocular diseases, including diabetic retinopathy (DR), retinal vein occlusion (RVO), age-related macular degeneration (AMD), retinopathy of prematurity (ROP), and so on [1]. Diabetic retinopathy and RVO, the two most common retinal vascular diseases, have been estimated to cause severe visual impairment in approximately 97 million people in 2012 [2] and 16.4 million people in 2008 worldwide [3]; however, the exact mechanisms of these remain unclear [1]. Clinically, early detection and diagnosis of retinal vascular diseases rely mainly on fundus fluorescein angiography, an invasive and expensive procedure. Moreover, the currently available therapeutic methods for retinal vascular diseases, such as laser photocoagulation, anti-vascular epithelial growth factor (anti-VEGF) or corticosteroid agent intravitreal injection, and vitrectomy, are limited by considerable disadvantages. These include high costs, the invasiveness of the procedures, adverse reactions following prolonged use, lack of preventative effects, and lack of suitability for advanced stages [4,5]. Therefore, breakthroughs though the development of new perspectives are desirable in order to understand the pathogenesis of retinal vascular diseases and to explore novel diagnostic and therapeutic strategies for these diseases.

With advances in high-throughput sequencing technologies, omics data (e.g., genomics, epigenomics, transcriptomics, proteomics, metabolomics, and microbiomics) have provided new opportunities for researchers to understand biological molecular processes through disease screening, diagnosis, staging, prognosis, and therapeutic response monitoring [6,7]. However, single omics data can only partially explain one aspect of the biological and molecular mechanisms underlying complex diseases. Multi-omics (multiple omics) provides an integrated perspective to power discovery across multiple levels of biology. This approach combines genomic data with data from other modalities, such as transcriptomics, epigenetics, and proteomics, to measure gene expression, gene regulation, and protein levels. It enables a combined, completed, complementary, and causative inference to be generated, which would contribute to an improved comprehension of disease studies and molecular studies [8,9,10,11].

In this review, the application of combined multi-omics data analysis to retinal vascular diseases is discussed in detail. First, we provide a general overview of the concept and practical value of multi-omics in retinal vascular diseases. We then summarize the multi-omics approaches implemented in the exploration of pathogenesis, diagnostic and prognostic biomarkers, and novel therapeutic strategies for retinal vascular diseases. Finally, the current challenges and future directions of multi-omics approaches are outlined.

## 2. Materials and Methods

### 2.1. Type of Omics Data

Recent advances in “omics” technologies have offered researchers unprecedented opportunities to study the correlation between biological molecules and complex diseases. Genomics, the first omics technology to appear, aims to identify specific genetic variants associated with disease states or biological processes, which could also provide invaluable insights to pinpoint causal variants [12,13,14]. Transcriptomics has been widely used to qualitatively and quantitatively detect RNA expression patterns in genome-wide specimens to provide information on gene expression, gene structure, and gene function in organisms [6,15]. Epigenomics investigates reversible modifications of DNA (DNA methylation) or histones at the whole-genome level, supplying regulatory information regarding chromatin accessibility and gene transcription [16,17]. Epitranscriptomics, also denoted as “RNA epigenetics,” is a novel emerging frontier field that can affect the fate of RNA through aberrant mRNA methylation modifications [18] such as 5-methylcytosine (m^5^C), N1-methyladenosine (m^1^A), N6,2′-O-dimethyladenosine (m^6^Am), and so on [19]. Another important omics concept is proteomics, a field that allows researchers to gain information pertaining to peptide abundance, post-translational modifications, and protein–protein interactions in order to qualify global protein changes [20]. In recent metabolomics studies, multiple metabolite molecules, including amino acids, fatty acids, carbohydrates, and other products of cellular metabolic functions, have been reported to play key roles in some metabolic syndromes such as diabetes, obesity, cardiovascular disease (CVD), and nonalcoholic fatty liver disease [21]. In contrast, spatial-omics profiles the native morphology of cells by detecting the cellular spatial context, cellular geometry, and extracellular matrix (ECM) around cells to probe for cell–cell or cell–ECM signaling interaction modalities [22]. Microbiomics and metagenomics are also common omics technologies used in clinical research to explore the correlations between disease status and environmental factors by analyzing the microbiota composition within a specific tissue [23,24,25]. Furthermore, the development of radiomics has enabled precision medicine to advance by providing invaluable imaging information such as the shape, size, and intensity of disease lesions in addition to data-characterization algorithms [26,27]. In addition, advances in single cell-omics can serve as a powerful tool for exploring the function of disease-associated cell types to understand the pathogenesis of complex diseases [28].

Each omics data type could provide useful information to delineate disease states or biological processes from different aspects, and the combined analysis of two or more types of omics data may contribute to generating a complementary and synthesized inference that supplies more knowledge than just the sum of each type of data. Combined analysis multiplies the discovery power and provides a more holistic view of biological systems by allowing researchers to witness the complicated interplay between the molecules of life.

### 2.2. Concept and Understanding of Multi-Omics Approaches

With the continually decreasing cost of high-throughput sequencing, increased grant funding for multi-omics studies, and advanced integration methods, multi-omics approaches are becoming increasingly feasible. Based on the central dogma, multi-omics datasets formulate the genetic information flow from DNA (genomics and epigenomics) to mRNA (transcriptomics) and eventually to protein (proteomics) and metabolite (metabolomics) aspects.

Over the past few decades, large-scale studies have focused on the molecular mechanisms underlying the procedures of understanding multifactor complex phenotypes. Combining genomics with proteomics or metabolomics can speed up the process by providing a complementary chain of evidence to connect genotypes (variants) to phenotypes (outcomes). Nevertheless, the genetic variants identified to date only explain a small part of certain phenotypes, and additional non-genetic factors, such as diet, lifestyle, and other environmental factors, likely in conjunction with susceptibility genes, commonly influence the pathogenesis of diseases. The integration of genomics, transcriptomics, and epigenomics can combine genetic predisposition and RNA regulation with environmental factors to comprehensively and intensively describe diseases, as illustrated in Figure 1.

Genomics is the main method used to identify disease-associated genetic variants, contributing to genetic risk prediction of different disease etiologies. GWAS are crucial genomic tools that enable researchers locate suspicious loci harboring causal variants; however, these can fail to directly implicate particular genes or pathways. In contrast, omics data-based trait analysis methods, such as expression quantitative trait loci (eQTL) analysis and fine-mapping, can integrate data originating from phenotype, genomics, and transcriptomics to identify causal genes and related pathways, supplying specific explanations for genetic factors in diseases.

The effects of non-genetic factors can be long-lasting and are sometimes heritable. Relative information can be obtained by discerning disease-associated modifications (a part of epigenomics), and these data may also serve as a bridge between genetic variants and the cumulative effects of environmental factors. In addition, the biological outcome of phenotypes can be determined through high-throughput proteomics and metabolite screening, from which the expression of proteins, amino acids, fatty acids, carbohydrates, or other products of cellular metabolism are intuitive reflections of the co-regulation of genetic and environmental factors.

In general, omics information derived from biological tissues in combination with radiomics and clinical information may delineate a holistic view for multi-omics approaches. Therefore, multi-omics allows researchers to directly measure the causes and consequences of biological phenotypes in addition to the regulated factors (RNA levels or environmental factors). This enables researchers to better connect genotypes to phenotypes, provides scientific insights that cannot be determined from single omics methods alone, and helps to fuel the discovery of novel drug discovery targets.

### 2.3. Multi-Omics Approaches in the Study of Retinal Vascular Diseases

Retinal vascular diseases are complex and multi-factorial in nature, and their clinical phenotypes can be influenced by genetic, epigenetic, environmental, and lifestyle factors. Therefore, understanding retinal vascular diseases based on single-level information obtained from only single-level omics data is inadequate. Multi-omics profiling studies enable a more comprehensive understanding of the molecular changes that contribute to normal development, cellular responses, and disease. In application, multi-omics approaches combine clinical data, omics data, and study design information to construct real-time high-digitization and high-precision analysis strategies that enable researchers to understand the pathogenesis, identify relevant diagnostic and prognostic biomarkers, and discover novel therapeutic strategies related to retinal vascular diseases.

### 2.4. The Characteristics of Retinal Vascular Diseases Require Multi-Omics Analysis

The retina is particularly vulnerable to vascular diseases owing to the high metabolic rate and high oxygen consumption of its photoreceptors. In addition to photoreceptors, retinal vascular diseases can also affect the function of intraocular immune cells, glial cells, pericytes, vascular endothelial cells, and the retinal pigment epithelium, which may lead to severe visual function loss through the development of ischemic zones and neovascular lesions [29,30,31]. It is crucial to investigate the function and response of intraocular cells throughout the course of retinal vascular disease to provide novel insights into these diseases. In this review, response variables were defined as disease-associated variants that can reflect the function and response of intraocular cells.

Omics technology can be used to identify global response variables through a single-factor perspective. However, multi-omics approaches allow for the delineation of panoramic views of retinal vascular diseases, as these can profile all possible response variables to ascertain the functional loss of intraocular cells and biological molecular processes during a disease’s course.

As an immune-privileged organ, the eye is highly shielded from systemic circulation due to unique anatomical and physiological barriers, such as the blood–retinal barrier, blood–aqueous barrier, and tight junctions within the conjunctival and corneal epithelium, which restrict macromolecules (e.g., cells, proteins, hormones, growth factors, lymphotoxins, and lipids) from the external and internal environment from entering ocular tissues [32]. Considering that all disease-associated response variables are restricted to a confined environment, intraocular vascular diseases are more likely to be clarified than other vascular diseases within the circulatory system, especially with the detection of intraocular specimens (e.g., aqueous, vitreous, and lesions from retinal vascular diseases). Multi-omics shows a powerful advantage in comprehending complex and active molecular biological processes in a confined environment, which corresponds to retinal vascular diseases in the intraocular environment. The main reason is that the comprehensive information obtained from multi-omics data can not only resolve the complexity of retinal vascular diseases but also provide a framework to identify all crucial disease-associated response variables in the intraocular environment.

Multi-omics research on retinal vascular diseases should be conducted with caution. For instance, human intraocular specimens should be collected by a trained vitreoretinal surgeon to avoid sudden drops in intraocular pressure, hemorrhage, and infection during the operation. For animals (e.g., mice, chickens, and rats), the integrity and freshness of the intraocular tissue should be ensured at the time of collection. In addition, it is necessary to design an optimal multi-omics scheme that can achieve the purpose of testing by using the minimum amount of specimen due to the limited total intraocular specimen volume that can be collected (e. g., the collected volumes of human aqueous and vitreous specimens are typically 0.1 mL and 0.3 mL, respectively). However, micro-detection and tremendous information flow derived from multi-omics approaches present new opportunities for understanding the molecular and clinical features of retinal vascular diseases.

## 3. Multi-Omics Approaches for Understanding the Pathogenesis of Retinal Vascular Diseases

It is challenging and valuable to elucidate the role of multi-omics in the pathogenesis of retinal vascular diseases owing to its multi-factorial properties.

### 3.1. Genetic Factors Explained through Multi-Omics Approaches

Advancements in multiple omics technologies and omics data-based trait analysis methods such as eQTL, genome-wide association studies (GWAS), transcriptome-wide association studies (TWAS), protein quantitative trait loci (pQTL), and epigenome-wide association studies (EWAS) have provided invaluable information to screen out novel pathogenic variants and dysregulated pathways associated with retinal vascular diseases. In addition, these allow researchers to comprehend the complex networks of multifactor interactions during the novel molecular mechanism exploration of these diseases. Over the past decade, GWAS have uncovered over 90 genetic variants at 55 loci associated with AMD, accounting for over 50% of the heritability of AMD [33,34,35,36,37,38,39,40].

In a recent multi-omics study, the genomic profiling of retinal pigment epithelium (RPE) samples differentiated from patient-induced pluripotent stem cells (IPSCs) identified 445 cis eQTLs related to AMD, and these were applied to RPE subpopulations [40]. Meanwhile, the transcriptomics and proteomics approaches in the aforementioned study also discovered some potential pathogenic pathways specified to regulate protein expression in geographic atrophy of AMD [40], including some related to mitochondrial functions (such as those associated with *PYROXD2* and *CRYZ*), metabolic functions (including lipid synthesis, gluconeogenesis, cholesterol synthesis, and glucose metabolism), and extracellular cellular matrix reorganization. The latter includes pathways associated with matrix metalloproteinases (MMPs; *MMP2* and *MMP16)*, tissue inhibitors of metalloproteinases (TIMPs) such as *TIMP2* and *TIMP3*, a disintegrin and metalloproteinase domain (ADAM; *ADAM10* and *ADAM12*), various cADAMs with thrombospondin motifs (ADAMTS), and five pQTLs (rs942813 in PYROXD2, rs989128 in spermatogenesis associated 20 (SPATA20), rs1424671 in CRYZ, rs2927608 in endoplasmic reticulum aminopeptidase 2 (ERAP2), and rs12041279 in recombinant ribonuclease L (RNASEL)) [40].

Few single-cell RNA sequencing studies have targeted some clusters of cells as critical modulators of retinal inflammatory responses and identified several vasoactive genes in DR, including angiotensinogen, a constituent of the renin–angiotensin system (RAS), as well as molecular pathways that adjust precise, metabolic, and oxidative stress-mediated changes [41,42]. In addition, a recent single-cell omics study explained the critical roles of neutrophils and neutrophil extracellular traps in tissue remodeling in retinopathy [43]. In addition, epigenomic evidence has shown that hypomethylated transcription factors (ETS1 and HES5) or proteins required for hematopoietic stem cell renewal and differentiation (PRDM16) regulate angiogenesis in diabetic retinopathy [44]. Moreover, an m6A-mRNA epitranscriptomic microarray was conducted to prove that fat mass and obesity-associated (FTO)-mediated m6A modification in amyloid-β(Aβ)-induced RPE degeneration may be a crucial pathogenic factor of AMD [45]. Another epitranscriptomic study profiled the mA methylation levels of mRNAs in retinas of oxygen-induced retinopathy (OIR) mice; their enrichment analysis showed that the hypermethylated mRNAs were involved in the dopaminergic synapse, glutamatergic synapse, and PI3K-Akt signaling pathways, whereas hypomethylated mRNAs were involved in autophagy, ubiquitin-mediated proteolysis, and spliceosome functions [46].

### 3.2. Non-Genetic Factors Illustrated through Multi-Omics Approaches

It is insufficient to describe the full picture of retinal vascular diseases from the perspective of genetic susceptibility. Environmental factors such as lifestyle have been found to change the progression of retinal vascular diseases. High-fat diets (HFDs) may enhance one’s vulnerability to ocular diseases by inducing alterations in the gut microbiota. The fecal microbiome and metabolome in a choroidal neovascularization (CNV) mouse model revealed altered microbial compositions, including *norank_f_Muribaculaceae*, *Candidatus_Saccharimonas, Prevotellaceae_NK3B31*, *Candidatus_Soleaferrea*, and *Truepera*. Altered levels of metabolites were also identified, of which 25 metabolites belonged to the superclass of lipids and lipid-like molecules, all of which may be associated with the pathogenesis of AMD [47]. Moreover, a growing number of studies based on advanced metabolomics technology [48,49,50,51,52] support a newly generated theory regarding the roles of adjusted metabolites (including fumaric acid, uridine, acetic acid, and cytidine) in diabetic retinopathy, and another combined omics analysis involving metabolomics and lipidomics identified several novel metabolites and lipid markers (including 2,4-dihydroxybutyric acid (DHBA), 3,4-DHBA, ribonic acid, ribitol, and triglycerides) associated with diabetic retinopathy [53].

Multi-omics analysis systemically explored candidate key factors from various aspects, including distinct genetic factors and some environmental functional factors, providing valuable evidence for probing the molecular mechanisms underlying complex multifactorial retinal vascular diseases. In addition, it is essential to comprehensively understand the molecular mechanisms of these diseases, as the classification and clear distinction of retinal vascular diseases according to their molecular mechanisms has the potential to improve patient outcomes by identifying and developing more targeted treatments.

## 4. Omics-Based Biomarker Discovery in Retinal Vascular Diseases

Omics data from eye biopsies can identify the molecular mechanisms of retinal vascular diseases in addition to the associated diagnostic prognostic and predictive biomarkers.

### 4.1. Current Potential Biomarkers for the Screening, Diagnosis, and Prognosis of Retinal Vascular Diseases

Currently, great challenges remain in the screening, diagnosis, and prognosis of retinal vascular diseases. In general, the screening of retinal vascular diseases depends on direct and indirect ophthalmoscopy or fundus photographic technologies. However, these methods are less likely to be carried out owing to the long annual screening period required for susceptible populations, emerging economic burden levels, and the scarcity of trained vitreoretinal specialists [54,55]. Although ophthalmoscopy might be sufficient for a basic diagnosis of AMD, fluorescein angiography (FFA) is the gold standard for detecting and differentiating these diseases, as this method allows clinicians to accurately determine the status and circulation of retinal vessels in order to uncover earlier diagnostic pathological indicators (e.g., microaneurysms, hemorrhages, small veins, and so on [56]), monitor disease progression through repeated examinations every 3–6 months for staging inference, and perform patient stratification through specific imaging findings. Nevertheless, its invasiveness and cumulative radiation toxicity limit its use in patients with cardiovascular and renal diseases, which are the most common comorbidities of retinal vascular diseases.

Severe proliferative vitreoretinopathy and neovascularization are common terminal changes in retinal and choroidal vascular diseases, and current therapeutic strategies have prolonged the time it takes patients to reach the terminal stages of these diseases. Nevertheless, a significant number of patients with poor treatment responses rapidly progress to the terminal stage within a short period. Presently, optical coherence tomography (OCT) and visual acuity are the main tools used to monitor the treatment responses and predict the prognosis of retinal and choroidal vascular diseases; however, inexperienced doctors are prone to generating inaccurate inferences through these methods. Therefore, there is an urgent need to discover more scientific and stable biomarkers for the screening, diagnosis, and prognosis of retinal vascular diseases.

Recent studies have reported potential biomarkers for resolving the problems stated above. Here, we discuss the roles of a classic biomarker for retinal vascular diseases, the vascular endothelial growth factor (VEGF) family. In retinal vascular diseases, retinal hypoxia and hypoperfusion trigger elevated levels of the VEGF family, including VEGF-A, VEGF-B, and placental growth factor (PlGF) in the intraocular environment, eventually leading to angiogenesis in the vascular and avascular zones, which is also well known as neovascularization [57,58,59]. As important stable biomarkers for the development stage and prognosis of retinal vascular diseases, it has been confirmed that a significant correlation exists between high levels of VEGF-A in aqueous and vitreous fluids and neovascularization of the retina, disc, angle, and iris in the proliferative DR (PDR) stage as well as in ischemic retinopathies and ischemic RVO [57,60,61].

Intravitreal injection of anti-VEGF agents is the first-line treatment strategy for ischemic retinal vascular diseases. Despite it being considered as the most promising biomarker, the VEGF family fails to comprehensively explain retinal vascular diseases. Other biomarkers, such as elevated levels of various cytokines, interleukin (IL)-1β, IL-2, IL-4, IL-5, IL-6, IL-10, interferon-γ, tumor necrosis factor-α, and VEGF, were observed in the aqueous humor or vitreous of patients with DR [62,63,64], AMD [65], RVO [66], and ROP [67]. However, all of these biomarkers were derived from a series of positive findings from faint single-aspect evidence sources, and further validation is necessary. Therefore, suitable and stable biomarkers for retinal vascular diseases remain elusive. Multi-omics approaches may simply be used to identify more precise, data-supported, stable, specific, and sensitive biomarkers with lower expenses, increased user-friendliness, and easier accessible detection methods by creating multi-dimensional global evidence chains from suitable tissues.

### 4.2. Omics-Based Biomarkers Discoveries

Newly-developed “omics-based biomarkers discovery” approaches may be potential solutions to the issues discussed above in previous sections, as these approaches have been frequently applied to screen out potential clinical detection indicators for cancers [68,69], type II diabetes [70], cardiac vascular diseases [71], central nervous system diseases [72], and autoimmune diseases [73]. In these approaches, as illustrated in Figure 2, omics data derived from biological specimen sequencing experiments or public databases are integrated together in an untargeted and unbiased manner (i.e., the “hypothesis-free method”) for further analyses. These methods have shown great promise in the identification of precise, specific, and sensitive candidate biomarkers for the clinical detection of retinal vascular diseases to reduce the prevalence of low vision and blindness.

The flow chart illustrates the omics data-based biomarker discovery process, including omics data collection, multi-omics data integration analysis, biomarker targeting, validation, and clinical application.

#### 4.2.1. Source of Omics Data

Regarding biological specimens for screening, early detection, and long-term monitoring of retinal vascular diseases, biomarkers in tears, saliva, sweat, and other systemic specimens have practical significance mainly due to the non-invasive collection processes and high concentrations of proteins and metabolites of these samples [74,75,76,77]. Moreover, if omics-based biomarker discovery is conducted using omics data from multiple studies, different types of omics data should be retrieved and downloaded from specified public databases, such as the recommended genomics databases (GWAS catalog: https://www.ebi.ac.uk/gwas/ (accessed on 24 December 2022) GWAS central: https://www.gwascentral.org/ (accessed on 24 December 2022) and dbGaP: https://dbgap.ncbi.nlm.nih.gov/ (accessed on 24 December 2022)); transcriptomics databases (GEO: https://www.ncbi.nlm.nih.gov/geo/ (accessed on 24 December 2022), Expression Atlas: https://www.ebi.ac.uk/gxa (accessed on 24 December 2022), and ArrayExpress: https://www.ebi.ac.uk/arrayexpress/ (accessed on 24 December 2022)); proteomics databases (Peptide Atlas: http://www.peptideatlas.org/ (accessed on 24 December 2022), ProteomicsDB: https://www.proteomicsdb.org/ (accessed on 24 December 2022), Human Proteome Map: http://www.humanproteomemap.org/ (accessed on 24 December 2022), and Human Proteome Atlas: http://v13.proteinatlas.org/ (accessed on 24 December 2022)); and metabolomics databases (Human Metabolome: https://hmdb.ca/ (accessed on 24 December 2022) and MetabolomeExpress: https://www.metabolome-express.org/ (accessed on 24 December 2022)).

#### 4.2.2. Multi-Omics Data Integration Analysis

After obtaining a variety of omics data, the critical step of multi-omics data integration is the identification and visualization of differentially expressed (DE) molecules, which can be accessed by auxiliary analysis platforms such as the Biomedical and Healthcare Data Discovery Index Ecosystem (bioCADDIE) [78] and Omics Discovery Index (OmicsDI) [79]. Next, the DE molecules are enriched in the biological functional analyses to generate significant pathways and relation networks (target relation or interaction relation) with the support of some advanced repositories or tools for enrichment analyses, including gene ontology (GO), gene set enrichment analysis (GSEA), Reactome, Kyoto Encyclopedia of Genes and Genomes (KEGG), protein–protein interaction (PPI) databases, and use of the Kinase Target Network and Transcription Factor Network. In the subsequent analyses, the key molecules with significant optimal biological functional (BF) annotations, critical locations in the related networks, and targeted molecular pathways can be determined through various eye and vision studies.

#### 4.2.3. Biomarker Targeting, Validation, and Clinical Detection

A prospective randomized controlled trial (PRCT) based on a small cohort in combination with special statistical approaches (e.g., Cox regression model, time-dependent receiver operating characteristic (ROC) curves, and Kaplan–Meier survival analyses) can be designed to estimate the prediction effectiveness, sensitivity, and specificity of potential biomarkers (the key output molecules from the previous step). Potential biomarkers with high sensitivity and specificity should be targeted as appropriate clinical indicators. However, the concrete mechanisms affecting retinal vascular diseases along with the stability, safety, and cost-effectiveness of candidates should be validated in vivo, in vitro, and in large-population PRCTs. Subsequently, some accurate, evidence-supported, stable, specific, and sensitive biomarkers with low costs and that can be detected using easily accessible detection methods can be applied and promoted in clinical screening, diagnosis, and prognosis determination.

### 4.3. Omics Biomarkers of Retinal Vascular Diseases

Recently, several omics studies have been conducted to identify potential biomarkers for understanding retinal vascular diseases. To deepen the practical value of the “omics-based biomarker discovery” approaches proposed in the previous section, we analyzed multiple omics data of retinal vascular diseases from specific data repositories. In the literature pertaining to omics data, some omics biomarkers for the screening, diagnosis, and prognosis of DR, AMD, RVO, and ROP were also determined.

#### 4.3.1. Omics Biomarkers in DR

Multiple omics biomarkers of DR can be determined through screening, diagnosis, and prognosis methods. In addition, details regarding the accession number, type, originated specimen, and concrete clinical application value of omics data were extracted from the corresponding literature, as shown in Appendix A.

Regarding DR screening, Guo et al. [80] designed a case-control study on serum metabolomics of 69 pairs of type 2 diabetic patients (T2DM) with and without DR, where thiamine and tryptophan metabolic disorders, downregulated trehalose, and upregulated choline and indole derivatives were targeted as potential metabolic biomarkers. In addition, in a retrospective case-control epigenomic study, two serum epigenetic biomarkers, namely cg02873163 and cg11343894 in S100A13, were used to stratify patients who developed DR with a DM duration of <3 years and those without DR for more than 30 years after being diagnosed with DM [81].

From the perspective of DR diagnosis, Skol et al. [82] combined insightful genomics and transcriptomics data with patient clinical phenotyping information (no diabetes, no diabetic retinopathy, and proliferative diabetic retinopathy) using summary data-based Mendelian randomization analysis. Their study found that genetic variation and elevated expression of the folliculin (*FLCN*) gene demonstrated a latent contribution to DR diagnosis and staging. In another study, integration analyses of metabolomic and lipidomic data revealed that four metabolic biomarkers, namely 2,4-dihydroxybutyric acid (DHBA), 3,4-DHBA, ribonic acid, and ribitol, and two lipid biomarkers, namely LDL cholesterol and triglycerides, were associated with different DR stages [53]. Moreover, proteomic biomarkers such as IL-2/-5/-18/-13, TNF, and MMP-2/-3/-9 in tears have been utilized in non-invasive strategies for the diagnosis and staging of DR in recent research [83].

In the case of DR prognosis, recent research has identified several diagnostic and prognostic miRNAs for recurrent vitreous hemorrhage in PDR patients through angiogenic miRNA profiling, including miRNA-19a and -27 as putative diagnostic vitreous biomarkers for PDR and elevated levels of miRNA-20a and -93 as prognostic biomarkers for recurrent hemorrhage incidence in PDR [84]. DR, a long-term progressive disease, necessitates frequent follow-ups and intervention throughout a patient’s lifespan. Therefore, with the intention of developing a non-invasive and reliable monitoring method, a study proposed that biomarkers screened out through metabolic and lipid phenotype evaluations of tear fluids and sweat (such as taurine) may be a promising tool in the prognostic assessment and monitoring treatment efficacy of diabetic retinopathy [85].

In addition to the above, researchers have emphasized the potential roles of metabolic biomarkers in the clinical diagnosis of DR. Li et al. [86] summarized and tabulated diagnostic and severity-associated metabolic biomarkers from pre-DR, NPDR, mild PDR, and PDR samples over the past 10 years, including pyruvate, Asp, glycerol, cholesterol, Cer (d18:1/24:0), ChE 20:3, and many other metabolites. In addition, Jian et al. [87] systemically reviewed the latest advances in metabolomics (from nuclear magnetic resonance (NMR) to liquid chromatography-mass spectrometry (LC-MS)) and the potential diagnostic metabolic biomarkers (12-hydroxyeicosatetraenoic acid (12-HETE), 2-piperidone, linoleic acid, nicotinuric acid, ornithine, and phenylacetylglutamine in human serum) from 39 original articles and then integrated the GWAS and metabolome data to predict molecular therapeutic targets (arginase 1, nitric oxide synthase 1, and phosphodiesterase). Furthermore, Hou et al. [88] summarized the findings of nine metabolomics studies and identified several potential diagnostic biomarkers, including L-glutamine, L-lactic acid, pyruvic acid, acetic acid, L-glutamic acid, D-glucose, L-alanine, L-threonine, citrulline, L-lysine, and succinic acid, as well as the significant metabolic pathways involved in amino acid and energy metabolism.

Radiomics biomarkers are highly promising for DR diagnosis and prognosis. Afarid et al. [89] analyzed the radiomics parameters in patients without clinical signs of DR compared to healthy individuals to determine several radiomic signs. Notably, enlarged foveal avascular zones (FAZs), reduced vascular density volumes in the macular area, and significant deviations of FAZ shape parameters (convexity and frequency domain irregularity) can be regarded as early screening identifiers of DR patients. Other Optical Coherence Tomography Angiography (OCTA) studies have also indicated that the size and shape of the FAZ may be correlated with a patient’s DR severity and grade [90,91]. Additionally, the retinal thickness measured by OCT B-scans [92] and retinal leakage index dynamics on ultra-widefield fluorescein angiography images [93] can be utilized to predict the response to intravitreal anti-VEGF treatment.

#### 4.3.2. Omics Biomarkers in AMD

Multi-omics data in combination with clinical phenotyping data could offer a vast amount of reliable digital indicators to advance research on AMD disease. Several omics, proteomics, metabolomics, and epigenomics technologies have been integrated to screen for biomarkers associated with AMD, and the results revealed that some compounds associated with the oxidative stress pathway, complement system, and lipid metabolism pathway were regarded as the most appropriate candidates [94]. In addition, the other omics biomarkers of AMD screening, diagnosis, and prognosis uncovered from omics research data from different databases are summarized in Appendix A.

In the context of AMD screening, several serum transcriptomic indicators, including hsa-let-7a-5p, hsa-let-7d-5p, hsa-miR-23a-3p, hsa-miR-301a-3p, hsa-miR-361-5p, hsa-miR-27b-3p, hsa-miR-874-3p, and hsa-miR-19b-1-5p, were more highly expressed in the serum of patients with dry AMD (*n* = 12) compared to wet AMD patients (*n* = 14) and healthy controls (*n* = 10), indicating that these dysregulated biomarkers may be potential diagnostic or screening tools for at-risk populations [95]. In a clinical serum metabolomics study [96], elevated serum concentrations of phosphatidylcholine, docoxahexaenoic acid, and eicosatetraenoic acid in choroidal neovascularization cases (CNV-AMD, *n* = 20) and polypoidal choroidal vasculopathy cases (PCV, *n* = 20) compared to healthy controls (*n* = 20) showed putative roles in identifying AMD patients among vulnerable populations.

Regarding AMD diagnosis, epigenomic research [97] has revealed that hypomethylation of the IL17RC promoter in AMD patients (*n* = 202) compared to controls (*n* = 96) can result in elevated concentrations of IL17RC proteins and messenger RNAs in the peripheral blood as well as in the affected retina and choroid. This suggests that the DNA methylation pattern and serum concentrations of IL17RC may be regarded as candidate biomarkers for the diagnosis of AMD. Another DNA methylation locus was recently reported, namely age-related maculopathy susceptibility 2 (ARMS2), and it has been shown that its CpG sites are significantly correlated with the risk SNP (rs10490924) genotype in these loci, thereby contributing to the development and progression of AMD [98]. Schori et al. [99] profiled proteins in the vitreous of AMD patients (dry AMD: *n* = 6; neovascular AMD: *n* = 10) using liquid chromatography coupled mass spectrometry. Specifically, upregulated cholinesterase (CHLE) in dry AMD patients and elevated ribonuclease (RNAS1) and serine carboxypeptidase (CPVL) in both types of AMD patients were identified as latent diagnostic biomarkers.

With regard to AMD prognosis and treatment response monitoring, the metabolites and proteins in systemic specimens may be suggestive of long-term benefits. Coronado et al. [100] designed a pilot prospect study to estimate the protein profiles of AMD patients with good responses to anti-VEGF agents (*n* = 5), poor responses to anti-VEGF agents (*n* = 5), and of healthy controls (*n* = 5). In their study, several dysregulated proteins were speculated to be potential biomarkers to distinguish AMD patients with different responses to anti-VEGF agents, and among these, LCN1 and the heavy chain component of IGHM were upregulated in the good response group and downregulated in the anti-VEGF poor response group. In addition, plasma kallikrein (KLKB1), kinase insert domain receptor (KDR), and vascular endothelial growth factor receptor 1 (VEGFR-1) were downregulated in the good response group and upregulated in the anti-VEGF poor response group.

The application of metabolomics has been highlighted in AMD research, and several reviews have summarized the potential metabolic biomarkers and pathways associated with AMD, emphasizing their roles in the construction of diagnostic or predictive models and new therapeutic targets for AMD [101,102,103]. Hou et al. [101] identified 108 dysregulated serum or plasma metabolites from the literature as potential candidate biomarkers and significantly enriched them in the adenosine, hypoxanthine, tyrosine, phenylalanine, creatine, citrate, carnitine, proline, and maltose pathways. Brown et al. [102] also summarized a list of metabolites from systemic samples from other studies, including phosphatidylcholine, C3 carnitine, 2-ω-carboxyethyl pyrrole (CEP), and lysoPE, proposing that biomarkers in alternative biofluids such as tears, saliva, and vitreous may be of practical value in the diagnosis, staging, and prognosis determination of AMD due to the blood–retinal barrier. In another study, Deng et al. [104] conducted untargeted metabolomics analyses of AMD samples and identified two metabolites that were differentially distributed between PCV and CNV. They found that the two metabolites, hyodeoxycholic acid and L-tryptophanamide, were differentially distributed between PCV and CNV, and the researchers investigated the genetic association of these metabolites with other metabolites in wet AMD. Liu et al. [105] revealed distinct metabolic pathways in wet AMD, PCV, and pathological myopia, in which serum metabolomes were profiled using gas chromatography coupled with time-of-flight mass spectrometry (GC-TOFMS). In addition, multivariate statistical methods and data mining were used to interpret the macular neovascularization of the samples. Personalized pathway dysregulation score measurement using the Lilikoi package in R revealed that the pentose phosphate pathway and mitochondrial electron transport chain were the most important pathways in AMD. Furthermore, purine metabolism and glycolysis were identified as the major disturbed pathways in PCV, whereas altered thiamine metabolism and purine metabolism may contribute to PM phenotypes.

Serum metabolomics is a powerful tool for characterizing metabolic disturbances in macular neovascular disease. Li et al. [86] summarized metabolomics-based clinical studies of AMD, and identified metabolic diagnostic and prognostic biomarkers/pathways in early, intermediate, and late AMD. Additionally, they reviewed some new therapeutic targets based on recent metabolomic findings, such as the pyruvate dehydrogenase kinase/lactate axis and findings related to taurine.

#### 4.3.3. Omics Biomarkers in RVO and ROP

After DR, RVO is the second most common retinal vascular disorder and can be divided into central retinal vein occlusion (CRVO) and branch retinal vein occlusion (BRVO) [3]. Proteomics research has revealed some significantly dysregulated molecules secondary to RVO. Yao et al. [106] profiled the proteins in the aqueous humor of patients with cataract controls and BRVO and found that several downregulated proteins, including fibroblast growth factor 4 precursor, hepatoma-derived growth factor isoform a, and α-crystallin A, may serve as biomarkers for the development of macular edema in RVO. A recent study reported several upregulated proteins in the retina of a laser-induced BRVO model, including laminin subunit β-2, laminin subunit γ-1, lipocalin-7, nidogen-2, osteopontin, integrin-β1, isoform 2 of α-actinin-1, and talin-2. This study also suggested that proteins involved in focal adhesion signaling and major extracellular matrix remodeling processes are associated with neovascularization and macular edema following BRVO [107]. In addition, other studies have illustrated that proteomic changes following aflibercept intervention in experimental CRVO [108] and BRVO models [109] can be applied to evaluate subjects’ responses to the intervention.

In their study, Xing et al. [110] conducted metabolomics analyses of the aqueous humor in patients with RVO and identified differential profiles of metabolites between mild and severe macula edema (ME). In severe RVO-ME, compared with mild RVO-ME, the degradation and biosynthesis of valine, leucine, and isoleucine; histidine metabolism; beta-alanine metabolism; and pantothenate and coenzyme A biosynthesis significantly differed. Furthermore, receiver operating characteristic (ROC) curve analysis revealed that adenosine, threonic acid, pyruvic acid, and pyro-L-glutaminyl-l-glutamine could differentiate RVO-ME from controls with an area under the curve (AUC) of >0.813. Urocanic acid, diethanolamine, 8-butanoylneosolaniol, niacinamide, paraldehyde, phytosphingosine, 4-aminobutyraldehyde, dihydrolipoate, and 1-(beta-D-ribofuranosyl)-1,4-dihydronicotinamide had an AUC >0.848 for distinguishing mild RVO-ME from severe RVO-ME.

Even with the best neonatal care and management strategies, ROP remains the major cause of blindness and visual disability in infancy [111]. Fortunately, advanced omics technology has provided new possibilities for the screening, diagnosis, prognosis, and management of ROP. For instance, Good et al. [112] utilized genomics to identify genetic factors to screen for potential treatment targets and explain the progression of ROP development. Furthermore, proteomics will serve as useful determinants of disease states and stages as well as diagnostic and prognostic tools for ROP. In another study, Danielsson et al. [113] profiled the serum proteins of 14 extremely preterm infants at postnatal days 1, 7, 14, and 28 and at postmenstrual ages of 32, 36, and 40 weeks, demonstrating that 11 proteins associated with ROP were potential biomarkers for predicting ROP. These eleven proteins were significantly enriched in angiogenesis, neurogenesis, osteogenesis, immune, and lipid metabolism functions and included AGER, ANGPT1, amyloid-beta precursor protein (APP), CD40 ligand (CD40LG), SLAM family member 5 (CD84), GDF2, heparin-binding EGF-like growth factor (HBEGF), macrophage metalloelastase (MMP12), peptidyl-prolyl cis–trans isomerase B (PPIB), plasminogen activator inhibitor 1 (SERPINE1), and thrombomodulin (THBD). Another proteomic study screened a total of 498 dysregulated proteins during vessel regression (PN12) and 345 dysregulated proteins during neovascularization (PN17) in the retinas of OIR pups that were involved in angiogenesis-related processes such as matrix remodeling, cell migration, adhesion, and proliferation. In addition, the mRNA expression in a human retinal endothelial cell (HREC) ROP model confirmed 56/69 neovascular-specific proteins, with 23 showing the same expression trends as OIR neovascular retinas. Finally, RNAi and transfection overexpression studies demonstrated that VASP and ECH1, which showed the highest levels in hypoxic HRECs, promoted human umbilical vein (HUVEC) and HREC cell proliferation, while SNX1 and CD109, which showed the lowest levels, inhibited their proliferation [114].

Zhou et al. [115] identified distinct profiles of plasma metabolites in patients with treatment-requiring ROP (TR-ROP) and controls. They also conducted metabolomic analyses in a murine ROP model and indicated that “protein digestion and absorption” and “aminoacyl-tRNA biosynthesis” were the most enriched pathways of the altered metabolites. These results demonstrated that the metabolomic profiles changed in the plasma of TR-ROP patients, and the altered metabolites could serve as potential biomarkers for the diagnosis and prognosis of TR-ROP patients. These findings illustrate that these proteins may act as potential biomarkers of vascular-related disorders. Moreover, a targeted blood metabolomic study reported dysregulated metabolites, including glycine, glutamate, leucine, serine, piperidine, valine, tryptophan, citrulline, malonyl carnitine (C3DC), and homocysteine, between 41 ROP and 40 non-ROP infants. In particular, high blood levels of C3DC and glycine can be promising screening biomarkers for ROP [116].

#### 4.3.4. Opportunities and Challenges That Coexist in Multi-Omics Integration Approaches Related to Biomarker Discoveries

Although a large number of omics biomarkers for the screening, diagnosis, and prognosis of retinal vascular diseases have been reported, there is still a lack of suitable candidates with strong evidence and stability for further validation and clinical detection, as these involve processes that require considerable manpower, financial costs, and time. Integration analysis of multi-omics data can optimize the biomarker discovery process by enhancing the evidence chain, minimizing the investment in manpower, materials, and time, and quantifying its stability in specimens.

The integration of distinct omics data and phenotypic information from previous studies should be a highly cost-effective method with which to discover candidate clinical indicators by enlarging the sample size, enhancing the statistical power, and shortening the validation process. In addition, the application of multi-omics methods in retinal vascular diseases is helpful for ophthalmic diseases as well as for systemic vascular diseases. As the only visible blood vessel in the human body, the early detection of retinal vascular lesions can indicate other systemic vascular diseases, and its lesions may be a representation of systemic vascular diseases. Through certain approaches using ophthalmological instruments, suspicious lesions can be screened for, and the root cause of the disease can be determined. In addition, the related changes in epigenetics, proteomics, and metabolomics can be unearthed through multi-omics methods. Finally, the metabolome may be a more accessible biomarker discovery means, as it can provide precise phenotype outcomes based on patient biology data.

The discovery of multi-omics biomarkers remains in its infancy, and great challenges exist in the associated practical application processes. The main challenges lie in the establishment or acquisition of detailed long-term clinical and phenotypic information datasets for further confirmatory studies, and this inevitably requires cooperation among multiple centers. Moreover, complex data integration requires exquisite programming technology and a solid foundation of clinical, genetic, and biological knowledge to discriminate key molecules from large sources of information. Therefore, the maturation of multi-omics biomarker discovery methods still requires a long-term development period.

## 5. Multi-Omics Applications in the Treatment of Retinal Vascular Diseases

### 5.1. Present Clinical Treatment Strategies for Retinal Vascular Diseases

Although the newly developed therapeutic strategy, anti-VEGF therapy, has been shown to be effective in reducing neovascularization and exudation to delay the progression of ischemic retinal vascular diseases, it does not restore visual function owing to its inability to rescue degenerating or dead retinal cells [117,118]. In addition, in some cases, prolonged anti-VEGF therapy may aggravate macular ischemia [119,120] and retinal cell apoptosis [121,122]. For the advanced phases of retinal vascular diseases, retinal laser photocoagulation can be conducted to seal the ischemic retinal non-perfusion zone other than the macula and destroy new retinal vessels to prevent severe vision-impaired complications, such as vitreous hemorrhage and retinal detachment [123]. However, some adverse effects limit its application, including aggravated macular edema, impaired peripheral visual field, nyctalopia, and reduced retinal sensitivity [124]. In addition, pars plana vitrectomy is the only option to improve the vision of patients with persistent vitreous hemorrhage (VH) or tractional retinal detachment (TRD), although a significant proportion of patients regain poor vision shortly after vitrectomy as a result of various reasons, including aggravated cataract, inflammatory irritated epiretinal membrane (ERM) formation, recurrent VH, and RD [125,126,127]. All these therapeutic options for retinal vascular diseases are far from satisfactory; thus, novel treatments are urgently required.

### 5.2. Multi-Omics Approaches in Identifying Novel Treatment Strategies

In this review, novel treatment measures developed based on multi-omics approaches will be described for two fields: first-in-class drug discovery and new applications of existing medicine.

#### 5.2.1. Multi-Omics Based First-in-Class Drug Discovery

Advanced omics technology has provided novel insights into the pathogenesis and development mechanism underlying retinal vascular diseases from molecular, genetic, and biological aspects; thus, numerous potential omics-derived treatment targets have emerged as potential drug candidates. Multi-omics evidence from different studies (e.g., transcriptomics [128], proteomics [129,130], metabolomics [131] and epigenomics [132]) can be integrated to support a novel gene therapy target for DR and AMD, the mammalian target of the rapamycin (mTOR) signaling pathway. This is frequently enriched to play a critical role in the pathogenesis of retinal degenerative diseases, including DR, AMD, ROP, and glaucoma [133,134,135,136]. Many studies have focused on this novel therapeutic target to design compounds for the intervention of retinal vascular diseases. Park et al. [137] manufactured an adeno-associated virus (rAAV)-delivered mTOR-inhibiting short hairpin RNA (shRNA) (rAAV-mTOR shRNA) to estimate the effect of mTOR inhibition on laser-induced CNV in mouse chorioretinal tissues, and the results showed substantial suppression of CNV, decreased local inflammation, and enhanced autophagic activity in CNV lesions. The same mTOR inhibitor, rAAV2-shmTOR-SD, was delivered into the vitreous of a streptozotocin-induced diabetic mouse model, and some indicators of DR progression, such as pericyte loss, acellular capillary formation, vascular permeability, and retinal cell layer thinning, seemed to be remarkably alleviated [138]. Another related treatment molecule for AMD, miR-24, can protect the retina from aberrant autophagy and RPE dysfunction by inhibiting the AKT/mTOR and ERK pathways by targeting chitinase-3-like protein 1 (CHI3L1) [139].

In addition to the above, other novel treatment targets originating from omics data findings also provide raw materials and research methods for multi-omics-based first-in-class drug discovery. Using the transcriptomics results of DR rats (*n* = 30) compared to controls (*n* = 45), it was recently found that THBS1 may be a key risk factor and a potential treatment target for DR. In addition, it was found that the expression of the THBS1 gene can be inactivated via the VEGF/Akt/PI3K pathway by silencing miR-20b-5p to inhibit DR [140]. In another study, Wang et al. analyzed the transcriptomic and cytokine array data of the retinal tissue of hyperglycemic mice with or without TNTL and found that macrophage-derived MIP-1γ may be regarded as a potential therapeutic target of botanical products TNTL in diabetic retinopathy [141].

In multi-omics-based first-in-class drug discovery, the important link for subsequent analysis after identifying key molecules or pathways is the determination of targeted novel compounds for treatment purposes. To acquire sufficient information for this process, we recommend potential drug target prediction databases such as PreDC (https://old.tcmsp-e.com/predc.php (accessed on 24 December 2022)), PharmMapper (http://lilab-ecust.cn/pharmmapper/submitfile.html (accessed on 24 December 2022)), PDTD (Potential Drug Target Discovery, http://www.dddc.ac.cn/pdtd/ (accessed on 24 December 2022)), and STITC (Search Tool for Interacting Chemicals, http://stitch.embl.de/ (accessed on 24 December 2022)).

#### 5.2.2. New Applications of Existing Medicine Based on Multi-Omics Findings

A significant proportion of key molecules or pathways involved in retinal vascular diseases have previously been delineated in other diseases (e.g., cancers or CNS diseases), and some mature targeted drugs have been promoted for clinical applications in the treatment of these diseases. In this section, we propose an existing drug discovery approach based on multi-omics findings in retinal vascular disease research.

Drug discovery databases can provide critical information regarding active target sites, biochemical functions, related signal regulatory pathways, and indications of drugs, such as TTD (Therapeutic Target Database, db.idrblab.net/ttd/ (accessed on 24 December 2022), DrugBank (http://www.drugbank.ca/ (accessed on 24 December 2022)), and ChEMBL (https://www.ebi.ac.uk/chembl/ (accessed on 24 December 2022)). Therefore, the main purpose of existing drug discovery approaches is to transfer key molecules or significant biological pathways derived from multi-omics research into these drug data repositories such that a list of potential candidate drugs for new applications can be identified.

In a literature-based drug discovery study [141], several raw omics data were retrieved from GEO and integrated together to mine DE genes that revealed significant biological pathways in rat models of diabetic retinopathy, including pathways related to regulation of cell proliferation, vasculature development, and negative regulation of macromolecule biosynthetic processes. The top three pharmacological compounds with the most upregulated genes associated with DE were dexamethasone, tretinoin, and ethanol, whereas the top three with the most enriched downregulated genes were ethanol, curcumin, and tretinoin. In addition, the top three genes with the highest reported frequency of inhibiting angiogenesis were statins, thalidomide, and angiostatins, whereas the top three associated with the highest reported frequency of disrupted cell proliferation were curcumin, tintinin, and resveratrol [142].

Multi-omics approaches can provide genetic and biological molecular evidence to scientifically guide the use of some existing drugs with unknown functional mechanisms. For instance, a few studies proposed that restoring one’s zinc balance can slow the progression of AMD [143,144,145]. In these studies, multi-omics approaches that combined transcriptome, proteome, and secretome analyses were implemented to determine the key molecular pathways in the zinc-induced changes in RPE, including pathways related to cell adhesion/polarity, extracellular matrix organization, protein processing/transport, and oxidative stress responses. In addition, a key upstream regulator effect similar to TGFB1 was identified [142].

Generally, multi-omics approaches can identify novel potential candidate drugs and guide scientific strategies for the treatment of retinal vascular diseases. In addition to developing new treatments, researchers are also committed to the strategy of combining pharmacovigilance data and omics data to evaluate the relationship between multi-omics factors and patients with low drug responses in order to identify personalized treatment biomarkers [146]. However, the identification of genetic variants, molecules, pathways, and networks remains a challenging endeavor that requires mathematical models, visualization, machine learning, data mining tools, and causal reasoning approaches combined with analyses to interpret and generate new, multi-evidence based, and precise intervention strategies [147,148].

## 6. Summary and Conclusions

The rise of multi-omics technology has given rise to advanced approaches for recognizing retinal vascular diseases. In this review, we reported the latest findings in retinal vascular disease research based on multi-omics technology and pinpointed the considerations that should be noted in disease-oriented exploration processes. Recent studies have identified molecules that are significantly associated with retinal vascular diseases, including the rs942813 genomic variant in AMD, the RLBP1, HIC1, and PARP12 genes in AMD, proteomic biomarkers such as IL-2/-5/-18/-13, TNF, and MMP-2/-3/-9, metabolomic variants such as C3DC and glycine in ROP, epigenomic variants such as hypomethylated ETS1 and HES5 in DR, and some altered microbes including *norank_f_Muribaculaceae, Prevotellaceae_NK3B31,* and *Candidatus_Saccharimonas* in CNV. Moreover, the significant pathways described in the recent literature of retinal vascular diseases include retinal inflammatory responses, metabolic and oxidative stress-mediated changes in DR, mitochondrial functions, metabolic pathways, extracellular cellular matrix reorganization in AMD, focal adhesion signaling and major extracellular matrix remodeling processes in BRVO, and matrix remodeling, cell migration, adhesion, and proliferation in ROP.

The intraocular environment is a special confined environment that is independent of the systemic circulation. This makes it possible for researchers to reproduce the intraocular biological environment through the construction of molecular interaction networks. In particular, with the advancement of single-cell sequencing technologies, multi-omics approaches at the single-cell level will allow researchers to correlate genetic variants and epigenetic effects with cellular processes and molecular functions in the biological environment. This thereby advances our understanding of the causes and consequences of biological phenotypes and allows researchers to generate systematic, multi-evidence-based inferences. In etiology and mechanism research, these inferences can provide new perspectives for existing hypotheses and help explore novel underlying molecular mechanisms. In addition, systemic and well-designed omics-based biomarkers and drug discovery methodologies have been proposed in our review to optimize the identification of screening, diagnostic, staging, or patient stratification-related prognostic biomarkers and the development of novel therapeutic strategies for retinal vascular diseases. Furthermore, combining targeted therapy with predictive biomarkers enables ophthalmologists to test the concepts of precision medicine, enabling trials to identify the right trial for the right patient at the right time. In summary, multi-omics technology provides great opportunities for personalized precision medicine.

Multi-omics has some limitations. First, truly complete multi-omics data should encompass all biological levels, developmental stages, cell types, ethnic backgrounds, and sexes. However, it is very difficult to obtain these data and identify the functional factors that mediate the control of core pathways and that drive gene expression. Second, the high costs, less accessible processes, complex computational integration analyses, and difficulties in the visualization, interpretation, and eventually generated inferences of these data limit their routine use in clinical and basic medical research. The inability to make broad, minimally biased measurements of a cell’s proteome is a major bottleneck for understanding how gene expression translates into a cellular phenotype. Particularly for proteomics and metabolomics, there are still many challenges in using proteomics to detect these markers and generalize them in clinical practice, including: (1) The complex pretreatment processes required, including the urgent need for the establishment of strict standard operating procedures. (2) The fact that multiple instrument platforms and ionization technologies coexist, making it difficult to develop a consistent procedure. (3) The fact that characteristic peptide analysis cannot fully represent the protein levels of a sample. Finally, (4) the fact that a complete data analysis strategy and a large number of samples are required for verification.

In the future, artificial intelligence (AI) technologies (e.g., machine learning (ML) and deep learning (DL)), modern molecular biotechnologies [149,150], and established systematic clinical information databases can be combined to perform multi-omics analyses, and these may allow clinical and basic medical researchers to effectively obtain inferences approaching precise personalized results associated with retinal vascular diseases.

## Figures and Tables

**Figure 1 cells-12-00103-f001:**
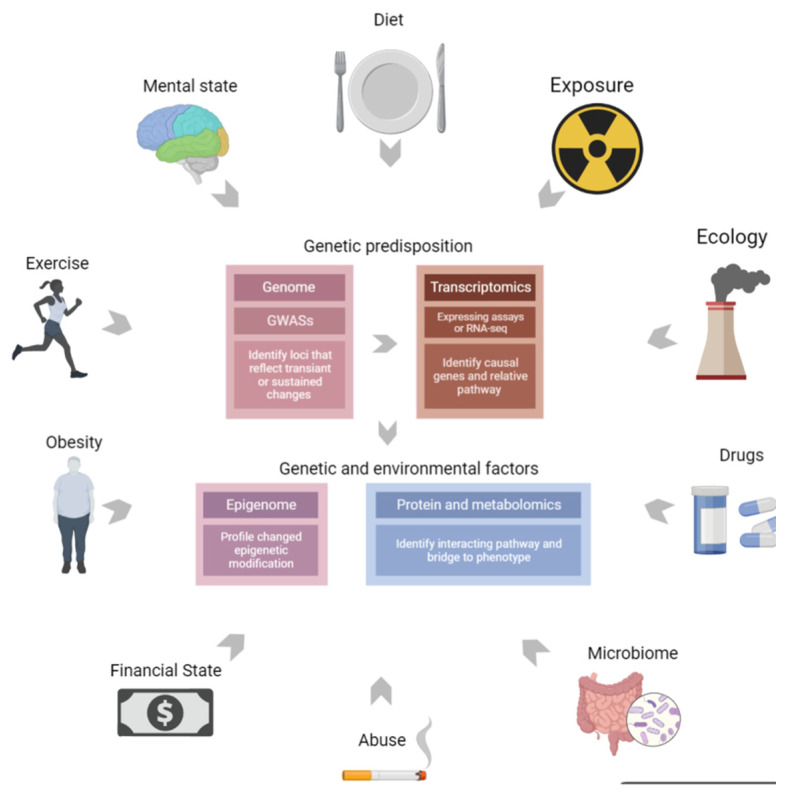
Multi-omics combine genetic predisposition with environment factors. Multi-omics can combine genetic predisposition, RNA regulation, and environmental exposure factors to describe diseases comprehensively and intensively.

**Figure 2 cells-12-00103-f002:**
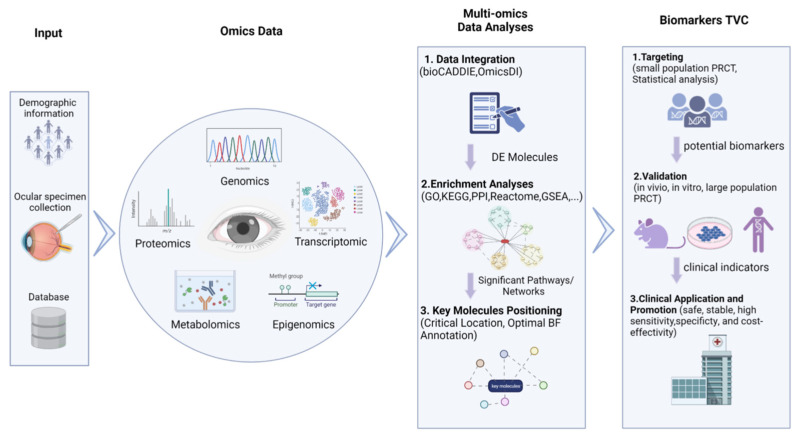
Multi-omics approach to identify biomarkers for retinal vascular diseases.

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
