# Peer review of "Essential Role of Multi-Omics Approaches in the Study of Retinal Vascular Diseases"

_cells, 2022, doi:10.3390/cells12010103_

Round 1

Reviewer 1 Report

Summary:

The roles of muti-omics approaches are described in characterizing retinal vascular diseases. This is a very timely and relevant topic that is useful to investigators focused on resolving the pathogenesis of retinal vascular diseases. However, some of the provided evidence requires clarification and supplementation.

1.     In addition to the omics approaches listed in this study, epitranscriptomics is another mode that must be included. One of its modes focuses on RNA m6A modifications. These studies have been instrumental in studying the pathogenesis of retinal vascular diseases. A Pub Med search using the search terms RNA m6A, epitranscriptomics, and retina generates 16 pertinent hits.

2.     Single-cell omics is emerging as one of the most powerful tools to unravel the complexity of retinal development and function. Its anticipated contribution to delineating retinal vascular disease pathogenesis warrants inclusion and discussion.

3.     Studies employing multi-omics approaches will help resolve the mechanisms underlying the regulation of life sustaining processes. There is a trend to combine the large data sets resulting from applying these different multi-omic approaches. This innovation has made significant strides in resolving the extreme complexity of underlying processes. Even though these different multi-omics approaches generate complex data sets, they remain tangled. This limitation complicates identifying the truly functional factors, mediating control of core pathway and driver gene expression. Although multi-omics is a promising model, the authors are encouraged to describe in detail the core technology that pinpoints the multi-omics approaches, which may be of interest to the reader. For example, please address:

1)    Identify and characterize the major remaining bottlenecks in proteomics or metabolomics studies

2)    Describe detailed limitations of multi-omics integration.

3)    Techniques need to be discussed that can simultaneously detect histone modification, of RNA expression, transcriptome, and proteome.

4)    Other shortcomings of multi-omics warrant consideration.

Author Response

Reviewer #1:

       The roles of muti-omics approaches are described in characterizing retinal vascular diseases. This is a very timely and relevant topic that is useful to investigators focused on resolving the pathogenesis of retinal vascular diseases. However, some of the provided evidence requires clarification and supplementation.

  1. In addition to the omics approaches listed in this study, epitranscriptomics is another mode that must be included. One of its modes focuses on RNA m6A modifications. These studies have been instrumental in studying the pathogenesis of retinal vascular diseases. A PubMed search using the search terms RNA m6A, epitranscriptomics, and retina generates 16 pertinent hits.

       Thank you for your valuable suggestion. According to your requirements, we have added relevant description of Epitranscriptomics in the revised manuscript: As suggested, we amended the suggestion, please see Page 2, Lines 66-69.

  1. Single-cell omics is emerging as one of the most powerful tools to unravel the complexity of retinal development and function. Its anticipated contribution to delineating retinal vascular disease pathogenesis warrants inclusion and discussion.

       We agree the reviewer’s opinion. We have added relevant description of Single-cell omics in the revised manuscript: As suggested, we amended the suggestion, please see Page 3, Lines 84-86.

  1. Studies employing multi-omics approaches will help resolve the mechanisms underlying the regulation of life sustaining processes. There is a trend to combine the large data sets resulting from applying these different multi-omics approaches. This innovation has made significant strides in resolving the extreme complexity of underlying processes. Even though these different multi-omics approaches generate complex data sets, they remain tangled. This limitation complicates identifying the truly functional factors, mediating control of core pathway and driver gene expression. Although multi-omics is a promising model, the authors are encouraged to describe in detail the core technology that pinpoints the multi-omics approaches, which may be of interest to the reader. For example, please address:

       We thank the reviewer for the favorable comments

  • Identify and characterize the major remaining bottlenecks in proteomics or metabolomics studies.

We agree the reviewer’s opinion. And we add the bottlenecks in proteomics or metabolomics studies,please see Page 18, Lines 779-787.

  • Describe detailed limitations of multi-omics integration.

We totally agree. And we detailed describe the limitations of multi-omics integration in Page 18, Lines 771-787.

  • Techniques need to be discussed that can simultaneously detect histone modification, of RNA expression, transcriptome, and proteome.

Thanks for the suggestion. To the best of our knowledge, there is currently no relevant technology that can be simultaneously detect histone modification, of RNA expression, transcriptome, and proteome.

4)    Other shortcomings of multi-omics warrant consideration.

       We totally agree. And we add this part in Page 18, Lines 771-787.

Reviewer 2 Report

This manuscript offers a comprehensive overview of multi-omics in retinal vascular diseases. The opportunities and challenges of multi-omics integration proposed by the authors are constructive and meaningful. However, there are some revision suggestions.

1. In this review, the applications of genomics, epigenomics, transcriptomics, proteomics, and metabolomics have been discussed in detail. But the emergence of radiomics was ignored. If there are some discussions about radiomics, the article will be more comprehensive.

2. Some sentences contain obvious grammatical mistakes. In line 35, the description of “the exact mechanism still remain incomplete understanding” is not appropriate. In line 54, “It enable a … inference generated” should be replaced by “It enables the generation of a … inference”.

Author Response

Reviewer #2

       This manuscript offers a comprehensive overview of multi-omics in retinal vascular diseases. The opportunities and challenges of multi-omics integration proposed by the authors are constructive and meaningful. However, there are some revision suggestions.

       We thank the reviewer for the favorable comments

  1. In this review, the applications of genomics, epigenomics, transcriptomics, proteomics, and metabolomics have been discussed in detail. But the emergence of radiomics was ignored. If there are some discussions about radiomics, the article will be more comprehensive.

      Thank you for your valuable suggestion. According to your requirements, we have added relevant description in the revised manuscript, please see Page 2, Lines 82-84.

  1. Some sentences contain obvious grammatical mistakes. In line 35, the description of “the exact mechanism still remain incomplete understanding” is not appropriate. In line 54, “It enable a … inference generated” should be replaced by “It enables the generation of a … inference”.

Thank you. We amended and the manuscript was edited by professional agent (www. Editage.com).

Reviewer 3 Report

The manuscript “The essential role of multi-omics approached in the study of retinal vascular diseases” well summarizes the recent fashion of omics in vascular diseases, including diabetic retinopathy, age-macular degeneration, retinal vein occlusion, and retinopathy of prematurity. However, I have a concern about some lack of interpretation.

1.     In chapter 3.3.2, they introduce several studies of the metabolome of AMD. Liu K and colleagues revealed distinct metabolic pathways among wet AMD, PCV, and pathological myopia (J Proteom Res., 19(2): 699-707, 2020). The authors should introduce it.

2.     Deng Y and colleagues conducted untargeted metabolomics of AMD and identified two metabolites that were differently distributed between PCV and CNV, hyodeoxycholic acid and L-tryptophanamide (Aging, 13(10):13968-14000, 2021). In addition, they also investigated the genetic association with metabolites in wet AMD. The authors should introduce it.

3.     Macular edema (ME) is a primary cause of severe impairment of central vision in RVO. Xing X and colleagues conducted metabolomics of aqueous humor in patients with RVO (Front Cell Dev Biol 9:762500, 2021). They identified differential profiles of metabolites between mild and severe ME. The author should describe it.

4.     Zhou Y and colleagues identified distinct profiles of metabolites from plasma between patients with treatment-requiring ROP (TR-ROP) and controls (Exp Eye Res., 2020). They also conducted metabolomics analyses in a murine ROP model. The authors should review them.

Author Response

Reviewer #3

       The manuscript “The essential role of multi-omics approached in the study of retinal vascular diseases” well summarizes the recent fashion of omics in vascular diseases, including diabetic retinopathy, age-macular degeneration, retinal vein occlusion, and retinopathy of prematurity. However, I have a concern about some lack of interpretation.

       We thank the reviewer for the favorable comments

  1. In chapter 3.3.2, they introduce several studies of the metabolome of AMD. Liu K and colleagues revealed distinct metabolic pathways among wet AMD, PCV, and pathological myopia (J Proteom Res., 19(2): 699-707, 2020). The authors should introduce it.

       As suggested, we introduced this research in Page 13, Lines 520-529.

  1. Deng Y and colleagues conducted untargeted metabolomics of AMD and identified two metabolites that were differently distributed between PCV and CNV, hyodeoxycholic acid and L-tryptophanamide (Aging, 13(10):13968-14000, 2021). In addition, they also investigated the genetic association with metabolites in wet AMD. The authors should introduce it.

       As suggested, we introduced this research in Page 12, Lines 515-520.

  1. Macular edema (ME) is a primary cause of severe impairment of central vision in RVO. Xing X and colleagues conducted metabolomics of aqueous humor in patients with RVO (Front Cell Dev Biol 9:762500, 2021). They identified differential profiles of metabolites between mild and severe ME. The author should describe it.

       As suggested, we introduced this research in Page 13, Lines 552-562.

  1. Zhou Y and colleagues identified distinct profiles of metabolites from plasma between patients with treatment-requiring ROP (TR-ROP) and controls (Exp Eye Res., 2020). They also conducted metabolomics analyses in a murine ROP model. The authors should review them.

As suggested, we introduced this research in Page 14, Lines 588-594.